# Technical Report on the New Ultrasound Lateral Mid-Shaft Approach to the Sciatic Nerve: A Never-Ending Story

**DOI:** 10.3390/medicina61010100

**Published:** 2025-01-10

**Authors:** Giuseppe Sepolvere, Mario Tedesco, Mario Cibelli, Dario Cirillo, Angelo Sparaco, Giuseppe Gagliardi, Giuseppina Costagliola, Loredana Cristiano, Valeria Rita Scialdone, Maria Rosaria Pasquariello, Fabrizio Di Zazzo, Luigi Merola, Mirco Della Valle, Daniela Arminio, Leonardo Maria Bottazzo, Marco Folliero, Giorgio Ranieri, Domenico Pietro Santonastaso, Antonio Coviello

**Affiliations:** 1Department of Anaesthesia, S. Michele Hospital, 81024 Maddaloni, Italy; giuseppesepolvere@gmail.com (G.S.); angelosparaco@gmail.com (A.S.); g.gagliardi@inwind.it (G.G.); pinacosta@alice.it (G.C.); lorecristiano@hotmail.it (L.C.); valeriascialdone79@gmail.com (V.R.S.); mariarosaria.pasquariello@gmail.com (M.R.P.); fab.dizazzo@gmail.com (F.D.Z.); luigimerola.lm@gmail.com (L.M.); mirco.dellavalle@virgilio.it (M.D.V.); danielaarminio@libero.it (D.A.); leonardomariabottazzo@gmail.com (L.M.B.); marcofolliero@yahoo.it (M.F.); 2Department of Anaesthesia, Mater Dei Hospital, 70125 Bari, Italy; mariotedesco@gmail.com; 3Department of Anaesthesia and Critical Care Medicine, University Hospitals Birmingham NHS Foundation Trust, Birmingham B15 2GW, UK; mario.cibelli@uhb.nhs.uk; 4Department of Neurosciences, Reproductive and Odontostomatological Sciences, University of Naples “Federico II”, 80131 Naples, Italy; antonio_coviello@live.it; 5Complex Operational Unit of Anesthesia and Operating Units, Department of Emergency and Internal Medicine, Isola Tiberina Hospital, Gemelli Isola, 00186 Rome, Italy; gioranieri84@gmail.com; 6Anesthesia and Intensive Care Unit, AUSL Romagna, M. Bufalini Hospital, 47521 Cesena, Italy; domenicopietro.santonastaso@auslromagna.it

**Keywords:** loco-regional anesthesia, nerve block, sciatic nerve, lower limb

## Abstract

The anatomy of the sciatic nerve allows it to be blocked at different levels using various anesthetic approaches. However, for several reasons, performing these approaches may be challenging or disadvantageous in specific categories of patients, particularly in obese patients. The objective of this brief technical report is to describe a new technical approach to sciatic nerve block, designed to simplify the procedure for certain categories of patients and less experienced practitioners. Since 2010, more than 5000 procedures have been performed by both experienced anesthesiologists and novice trainees in several hospitals. The ultrasound lateral mid-shaft technique appears to be a safe and effective method for performing a sciatic nerve block, even in obese patients with significant subcutaneous fat and unclear ultrasound images. This approach is particularly beneficial given the various anatomical variations that can occur. By targeting the mid-thigh area, the ultrasound beam accesses anatomical structures that are more superficial, improving the technique’s efficacy. Various hospital groups have been performing this technique as a routine procedure, achieving a success rate of nearly 100%. This impressive success rate exceeds that of other conventional techniques documented in the literature. Additionally, there have been significant improvements in comfort and ease for anesthetists. This method allows the anesthetic to spread around the paraneural sheath, covering the posterior femoral cutaneous nerve. Finally, it is performed in the supine position without the need to mobilize the lower limbs, ensuring patient comfort, especially in cases of fractures or lower limb injuries. Further studies are needed to confirm these results.

## 1. Introduction

The ultrasound-guided sciatic nerve (SN) block is a crucial anesthetic technique used in lower limb surgery [1]. It can serve as a single anesthesiologic method or be combined with other peripheral nerve blocks for procedures involving the femur, knee, tibia, fibula, and foot, including arthroscopic, prosthetic, and trauma surgeries [2]. The SN originates from the lumbosacral plexus and consists of fibers from the L4, L5, S1, S2, and S3 nerve roots [3]. From the pelvis, it enters the gluteal region through the ischiatic foramen, passing below the piriformis muscle. In the thigh, it runs close to the linea aspera of the femur until it reaches the apex of the popliteal fossa, where it terminates and divides into two main branches: the common peroneal nerve (CPN) and the tibial nerve (TN), which are surrounded by a layer of connective tissue called the paraneural sheath [3,4,5]. This bifurcation typically occurs at the apex of the popliteal fossa, though there is significant anatomical variability, with implications that are important in clinical practice when performing popliteal SN block [6]. Due to the anatomical characteristics of the SN, it can be blocked at various levels and using different techniques to achieve the desired anesthesia and analgesia [2]. However, these approaches may be challenging to implement for all patients due to several reasons [7]. The popliteal and original lateral approaches require knee flexion to be performed, which means they cannot be easily used on patients with an inability to flex the knee or fractures of the lower limb that produce an uncomfortable condition for the patient [8,9]. The anterior approach can be challenging because the nerve target is located deep within the tissue [10]. The lateral approach, described by Yoshida et al., is an alternative to the subgluteal approach and may be difficult to perform in obese patients [7]. Finally, in obese patients, all of the approaches discussed can be challenging to perform due to their Body Mass Index (BMI) being greater than 35 kg/m^2^ and the potential for unclear ultrasound images [7]. The purpose of this brief report is to describe in detail a new lateral approach technique for SN block, which aims to simplify the procedure for specific categories of patients and for less experienced practitioners.

## 2. Materials and Methods

This article provides a detailed description of the technical approach previously presented in a letter to the editor by Tedesco and colleagues in 2019 [11].

Since its initial proposal by S. Michele Hospital in Maddaloni, Italy, Mater Dei Hospital in Bari, Italy, Federico II University in Naples, Italy, and University Hospitals Birmingham, UK, more than 5000 procedures have been performed both by experienced anesthesiologists and novice trainees since 2010.

### 2.1. Technique Description

The procedure is conducted with the patient in a supine position, requiring no mobilization of the lower limbs. A 3–5 MHz convex ultrasound probe is positioned at the mid-tight in the muscular groove formed by the Vastus Lateralis (VLM) and Biceps Femoris (BFM) muscles, with the projection of a lateral to medial ultrasound beam (Figure 1).

The resulting lateral–medial planar ultrasound section reveals the position of the main body of the SN at the mid-shaft level of the femur, approximately 4–5 cm deep and laterally from the skin. The nerve is surrounded superiorly by the VLM, laterally by the BFM, and inferiorly by the semitendinosus (STM), semimembranosus (SMM) and adductor magnus (AMM) muscles (Appendix A).

As a result, this approach provides a better ultrasound resolution with a lower nerve target depth and a better chance of directing the needle to the target, providing less stressful conditions, even in patients with a high BMI.

The focus adjustment is performed by moving the marker on the side of the monitor to the desired depth of focus. Typically, the depth of focus is about 7 cm, or 0.5 cm more in front of the target nerve [12]. It is crucial that the ultrasonic beam is positioned perpendicular to the nerve due to its anisotropic behavior. Partial anesthesia can occur because the sciatic nerve bifurcates into the tibial and peroneal branches at the level of the infrapiriform foramen in approximately 11% of patients [13].

Bifurcations of the sciatic nerve in the proximal half of the thighs occur in 3.6% of limbs in cadaver studies. Interestingly, the sciatic nerve may bifurcate at different levels in both thighs of the same patient. This observation indicates that the sciatic nerve can divide into its terminal branches at any level within the thigh [14].

Performing a cranial and caudal ultrasound scan before the procedure allows the anesthetist to identify this division, ensuring that both branches are effectively blocked during the procedure.

After the careful disinfection of the skin of the lateral region of the mid-thigh, in a lateral to medial direction, an 80–100 mm short-bevel needle is then advanced in-plane. Its correct position is double-checked through the progressive injection of 3 mL of saline solution, and 20 mL of local anesthetic is deposited to form a halo of fluid after passing the paraneural sheath to surround the nerve visualized in the transverse section. When the paraneural sheath envelops the two branches separately, the CPN and the TN are visualized also at the mid-shaft thigh level (Figure 2A). As a final check, we routinely open the long-axis view of the nerve to ensure that the spread of local anesthetic has extended caudad and rostral; in particular, the cranial local anesthetic spread is assessed up to the subgluteal region as confirmation of the successful block of the posterior femoro-cutaneous branch (Figure 2B).

### 2.2. Real-Life Example Case

The effectiveness of the technique has been proven in several complex clinical scenarios. We describe the case of a 78-year-old patient affected by diabetes mellitus, arterial hypertension, severe chronic pulmonary disease, peripheral and cerebral vasculopathy, and chronic myocardial ischemia under clopidogrel therapy; this patient was also suffering from acute thrombosis of the popliteal artery and scheduled for urgent thromboendarterectomy surgery. The patient complained of acute severe lower limb pain from the popliteal region down to the foot; this was refractory to morphine administration, with cyanosis and marked ischemia. Due to anticoagulation and the respiratory profiles, neither spinal nor general anesthesia were considered as anesthetic strategies. We opted for a combination of femoral, obturator, lateral cutaneous femoral and lateral mid-shaft sciatic nerve blocks in order to avoid further complications deriving both from the anticoagulation status and orotracheal intubation. In addition, the mid-shaft sciatic nerve block was useful in the case of an 81-year-old patient who was affected by senile dementia, bilateral carotid artery stenosis, severe chronic pulmonary disease and severe aortic valve stenosis; this patient was on therapy with ticlopidine and scheduled for unstable pertrochanteric femoral fracture surgery. With light sedation and spontaneous breathing and without patient mobilization, we performed a combination of femoral, obturator, lateral cutaneous femoral and lateral mid-shaft sciatic nerve blocks involving the PFCN region through the cranial spread of LA, avoiding adverse effects related to general and spinal anesthesia. Moreover, the procedure was successfully used for lower limb amputation, starting from the mid-shaft region, in a 73-year-old patient admitted to the intensive care unit and affected by previous cerebral stroke, peripheral vascular disease, left carotid artery and mitral valve stenosis, atrial fibrillation, and acute pneumoniae; this patient was on therapy with intravenous antibiotics and warfarin. Their lower limb was necrotic and showed the initial signs of systemic infection. With spontaneous breathing and light sedation, we chose to perform a combination of femoral, obturator, lateral cutaneous femoral and lateral mid-shaft sciatic nerve blocks, allowing amputation in a critically ill patient. Finally, this approach was deemed safe and effective for a 67-year-old patient with hypertension who had sustained a compound fracture of the atlas and traumatic injury of the right knee after accidentally falling from a ladder. This patient was a candidate for open knee surgery. The fracture of the atlas was stabilized with a rigid collar, which made general anesthesia impractical due to the patient’s inability to extend his neck. Additionally, performing spinal anesthesia was a high-risk option because of the risk of spinal cord injury and the need to mobilize the patient to perform the anesthesiologic procedure (Appendix A). No complications or adverse effects were recorded for the four patients.

The key steps of this technical approach are summarized in Figure 3.

## 3. Discussion

In this technical report, we present a new approach to performing SN block that appears to be a feasible, effective and safe alternative to the well-described and established techniques. The new lateral ultrasound approach to the SN at the mid-thigh level offers several advantages: it eliminates the need to mobilize both the patient and the lower limb, enhancing comfort. In addition, it targets a more superficial nerve, making the procedure easier to perform and providing a quicker learning curve, even for those with less experience. Additionally, there are no major procedural challenges, such as the need to apply significant pressure on the probe to better visualize the anatomical structures, which results in the hand of the anesthetist becoming tired and the loss of images.

The popliteal approach is limited by significant anatomical variability; in fact, in 53.33% of cases, the SN divides near the upper angle of the popliteal fossa; in 26.66% of cases, the division occurs in the middle of the posterior thigh, while 13.33% of cases show the division at the upper one-third of the posterior thigh; only in 6.66% of cases does the CPN pass through the piriformis muscle, with the TN situated below the piriformis muscle [15]. Moreover, the popliteal approach requires the patient to be in prone position, rendering the procedure uncomfortable for those patients affected by lower limb fractures (Figure 4A) [2].The original lateral approach requires the patient to adopt a supine position with the leg bent, a lateral to medial needle direction, and the probe placed below the thigh with a posterior to anterior ultrasound beam used, similar to the popliteal approach performed with the patient in a supine position. The challenges consist of the uncomfortable position adopted by both the patient, unable to flex their leg, and the anesthetist (Figure 4B) [7,9,11].In the anterior approach to the SN, the patient is placed in a comfortable supine position, but the nervous target is overly deep, making it difficult for the local anesthetic to surround the nervous target and making the procedure painful (Figure 4C) [2,7,10,11].In the subgluteal approach, the patient is placed in a lateral position with the lower limb upward. Similar to the popliteal approach, the procedure could be uncomfortable in the case of fracture (Figure 4D) [2,7,11].

**Figure 4 medicina-61-00100-f004:**
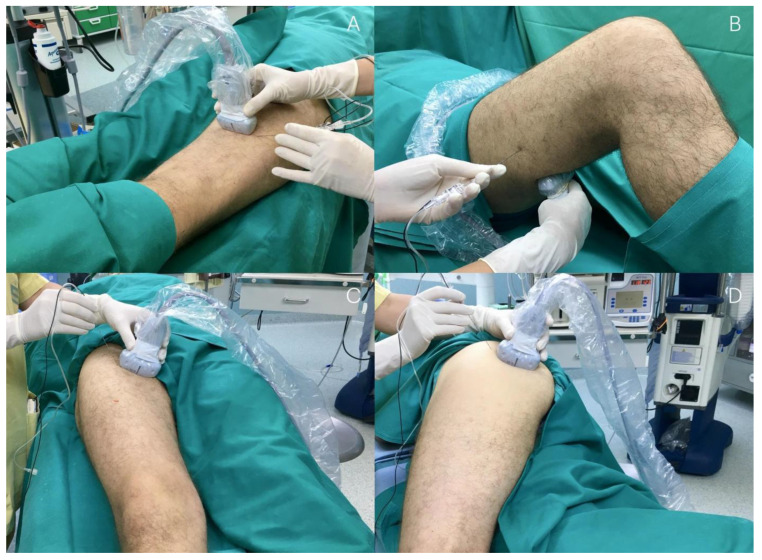
Different approaches to the sciatic nerve. (**A**) Popliteal approach: when performing the block, the prone position is uncomfortable in the case of lower limb fracture. (**B**) Lateral approach: the original lateral approach requires the patient to adopt a supine patient with their leg bent, a lateral to medial needle direction, and the probe placed below the thigh with a posterior to anterior ultrasound beam, similar to the popliteal approach performed with the patient in a supine position. The challenges consist of the uncomfortable position adopted by both the patient, unable to flex their leg, and the anesthetist. (**C**) Anterior approach: the patient is placed in a comfortable supine position, but the nervous target is overly deep, making it difficult for the local anesthetic to surround the nervous target by the local anesthetic and making the procedure painful. (**D**) Subgluteal approach: the patient is placed in a lateral position with the lower limb upward. Similar to the popliteal approach, the procedure can be uncomfortable in the case of fracture.

We appreciate and congratulate Yoshida et al. for describing a valid and effective alternative to the subgluteal approach, the original lateral approach and the anterior approach. However, it seems to be limited by the skin–nerve distance, which results in a poor ultrasonographic resolution and requires the needle to travel a long distance between tissues in the gluteal region before reaching the target, making it more difficult for inexperienced operators and more difficult to perform in obese patients, who have a more conspicuous subcutaneous tissue layer [7,11]. Similar to the lateral approach proposed by Yoshida, our technique provides better anatomical ultrasound landmark identification, which is more superficial in the thigh compared to the subgluteal region, and better needle visualization due to the improved angle alignment between the ultrasound beam and the needle, unlike the anterior approach and the original classical approach [7,10,16,17]. Similar to the classical lateral approach and the lateral approach proposed by Yoshida, the approach we propose involves the posterior femoral cutaneous nerve (PFCN) [2,7,11]. The PFCN (S1–S3) runs in the gluteal and thigh region near the SN until it separates from it by laying dorsal to the long head of the BFM [2]. The local anesthetic solution injected below the paraneural sheath surrounding the SN spreads significantly both cranially and caudally, involving the PFCN and allowing analgesia of the posterior tight, a crucial detail for lower limb surgeries (see Figure 3) [7,18]. Over the years, no patients have reported pain in the anatomical region covered by the PFCN. In addition, the approach we described, unlike the anterior approach, which is more frequently associated with increased pain in the posterior region of the thigh [7,9,19], has been effectively and safely used in combination with anterior nerve blocks in several cases of above-knee amputation.

### Limitations and Technical Consideration

After more than 15 years and 5000 procedures performed in four different hospitals, the described technique appears to be safe and effective, with a low rate of complications, among which Local Anesthetic Syndrome Toxicity (LAST) [8] and intraneural and intravascular injections are the most common. At that anatomical level, in a few cases the presence of a vessel, generally an artery, within the body of the nerve can be described (Figure 5).

In this condition, caution should be exercised while performing the block in order to avoid both the intraneural and intravascular injection potentially producing neural injury and hematoma [20].

To prevent intravascular injection, various practices have been attempted. Aspiration before injection is not completely safe because a negative aspiration can only be achieved with vessels of a good caliber and in patients who have an adequate blood volume. A good aspiration technique involves aspirating every five milliliters of solution injected or every time the needle’s tip is repositioned [21]. To prevent intraneural injections, several methods are currently in use, such as the paresthesia technique, using a peripheral nerve stimulator, the RAJ test, and assessing the injection pressure [21].

As described, through the lateral mid-shaft approach, the anesthetic solution may spread up to the PFCN following the paraneural path. The sheath surrounding the SN was found to be thin, transparent, and fragile, enveloping the nerve as a structure distinct from the epineurium [22]. The anatomy of the SN is more complex than previously described. The TN and CPN within the SN trunk appear to be centrally separated by the Compton–Cruveilhier septum and encompassed by their own paraneural sheaths. This unique internal architecture of the SN appears to promote the proximal spread of local anesthetic to the internal aspect of the SN trunk after a subparaneural injection at or below the divergence of the TN and CPN [23]. On this basis, the injection of local aesthetic below the sheath appears to be mandatory for the success of the block. When the needle passes the sheath, the tip may enter the nerve, increasing the risk of intraneural injection. The use of saline solution, before the injection of the anesthetic solution, may be useful to better identify the anatomical space and correctly place the needle tip.

In some particular cases, some patient-specific anatomical challenges may occur, rendering the technique more difficult to perform. Patients may show a lack of ultrasound landmark imaging if suffering from peripheral vascular diseases, or in a state of dehydration, or in the case of muscular hypotrophy, such as in elderly patients. In addition, in cases of significant muscle tone and mass, such as in athletic subjects or patients with morbid obesity, the movement of the body and the needle tip may be difficult when aiming to surround the nervous target with the local anesthetic solution. These anatomical conditions may also occur for the established approach to the sciatic nerve, and we believe that there are no further situations where other approaches may still be preferable. The patient never changes their supine position, as in the case of the anterior approach or unlike the popliteal one; the mid-shaft approach involves the PFCN, as in the case of the subgluteal approach, leaving the patient supine. Our approach is fully lateral in terms of the needle direction and ultrasound beam through a slight flexion of the leg, unlike the classic lateral approach to the sciatic nerve.

Our routine procedure achieved an impressive success rate of nearly 100%. This success rate surpasses that of other conventional techniques reported in the literature. Additionally, our technique offers significant improvements in comfort and ease for anesthetists.

However, it is important to note that our method has limitations, including the lack of comparison with the success rates and complications associated with other established approaches. The report is based on retrospective observations and lacks a strong prospective comparison. Although it mentions over 5000 procedures, it does not provide detailed data on patient demographics, variations in BMI, or specific clinical situations. Risks such as intraneural or intravascular injections are not analyzed quantitatively or compared with other techniques. Additionally, the performance of the reported technique across various clinical settings and among operators with differing levels of expertise is not adequately addressed.

Future research could evaluate the long-term outcomes of this approach. Additionally, a comparison of the effectiveness of this method with other traditional approaches could be suggested in a multicenter randomized clinical trial.

## 4. Conclusions

The ultrasound lateral mid-shaft technique offers a new approach to the SN that is both effective and safe, even for obese patients. This is because the ultrasound beam travels through the mid-thigh, where anatomical structures are more superficial and visible compared to the gluteal region, making the procedure easier to perform. Additionally, it provides a quicker learning curve for those with less experience, as there is no need to apply significant pressure on the probe to better visualize the anatomical structures. Furthermore, the injection is delivered below the paraneural sheath, allowing for the cranial and caudal spread of the local anesthetic solution that covers the posterior femoral cutaneous nerve (PFCN). Importantly, the procedure is conducted while the patient is in a supine position, and no lower limb mobilization is required. This enhances patient comfort, particularly in cases of fractures or lower limb injuries. However, further studies are necessary to validate this technique through multicenter research.

## Figures and Tables

**Figure 1 medicina-61-00100-f001:**
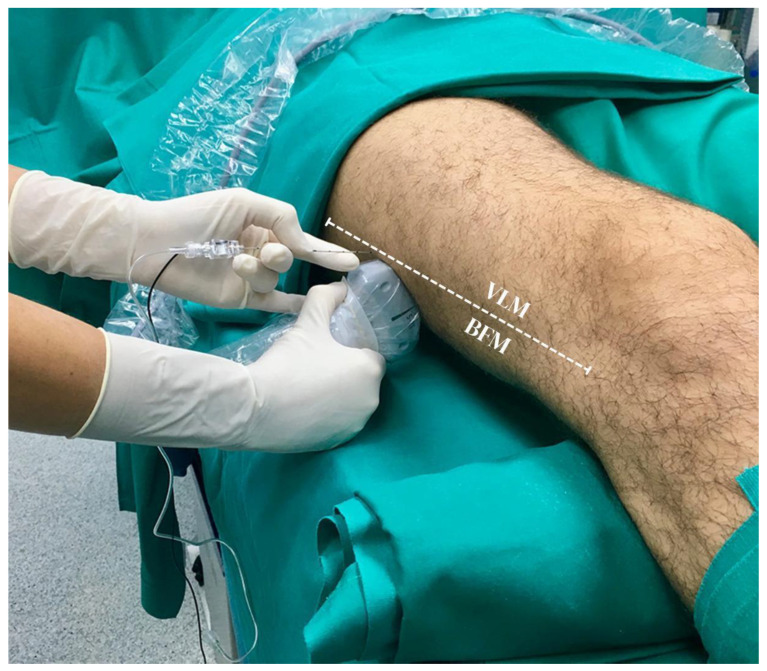
A convex ultrasound probe is positioned at the mid-tight in the muscular groove formed by the vastus lateralis and femoral biceps muscles (white dotted line), with the beam oriented upward. VLM: vastus lateralis muscle; BFM: biceps femoris muscle.

**Figure 2 medicina-61-00100-f002:**
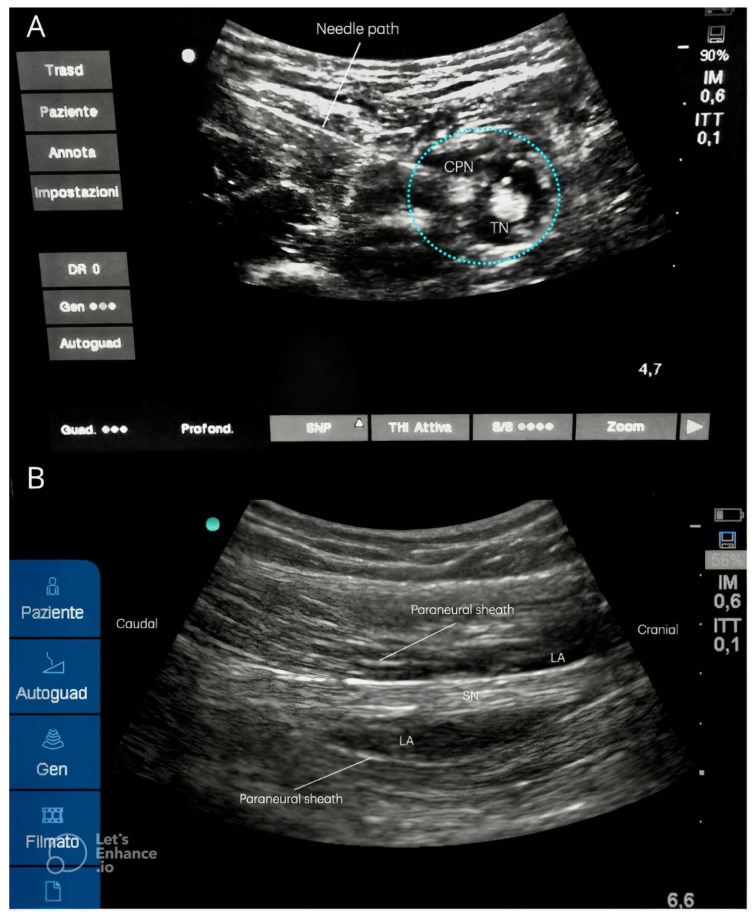
(**A**) Sciatic nerve divided into its components within the same paraneural sheath (light blue dotted circle) after the injection of local anesthetic. CPN: common peroneal nerve; TN: tibial nerve (**B**) Ultrasound long-axis view after lateral mid-shaft sciatic nerve block: the local anesthetic correctly injected below the paraneural sheath may spread for several centimeters in a caudal to cranial direction up to the subgluteal region covering the posterior femoro-cutaneous branch. LA: local anesthetic; SN: sciatic nerve.

**Figure 3 medicina-61-00100-f003:**
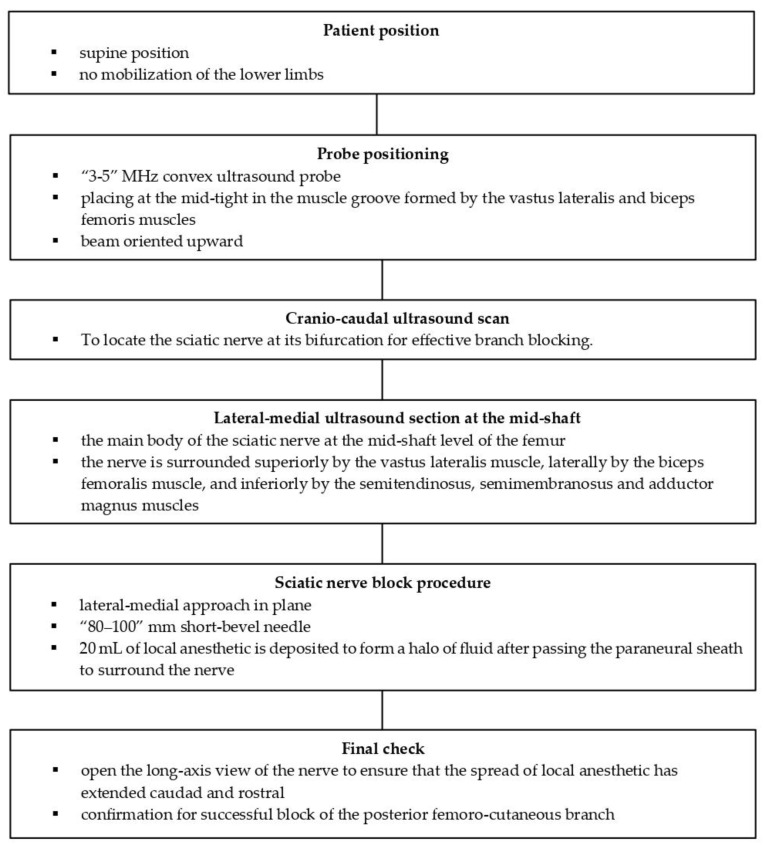
New ultrasound lateral mid-shaft approach to the SN: step by step flowchart.

**Figure 5 medicina-61-00100-f005:**
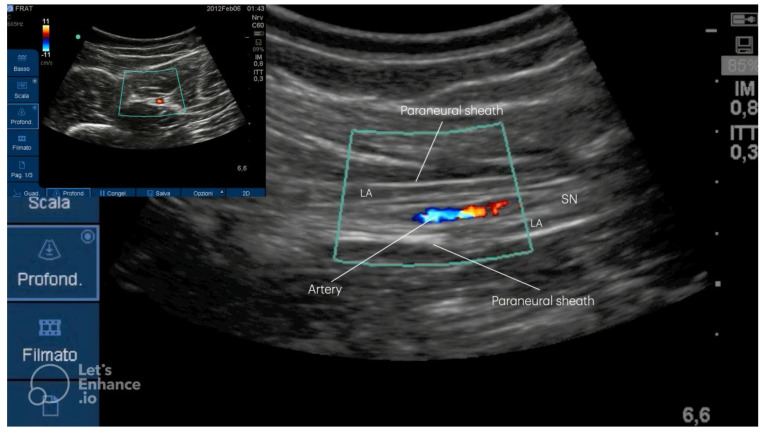
The presence of a vascular structure could be visualized within the body of the sciatic nerve. In the picture, the doppler shows the presence of the artery on the short axis inside the body of the sciatic nerve. SN: sciatic nerve; LA: local anesthetic.

## Data Availability

No new data were created or analyzed in this study. Data sharing is not applicable to this article.

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
