# Peer review of "Technical Report on the New Ultrasound Lateral Mid-Shaft Approach to the Sciatic Nerve: A Never-Ending Story"

_medicina, 2025, doi:10.3390/medicina61010100_

Round 1

Reviewer 1 Report

Comments and Suggestions for Authors

TITLE

  • Could you add the study type?

 Narrative Structure

  1. Missed or Misinterpreted Trends or Patterns:
    • Lines 27-36: Include a discussion on variability of nerve depth based on patient BMI and anatomical differences.
    • Lines 103-132: Add real-life case examples to demonstrate the successful application of the technique in complex scenarios.
    • Lines 181-236: Emphasize the preventative measures and potential challenges related to intraneural injections during the thoracic outlet syndrome (TOS) block.
  2. Under or Over-Emphasized Results:
    • Lines 33-36: Provide context on how this new approach compares with existing methods in terms of efficacy and safety.
    • Lines 74-102: Balance the discussion by including more detailed evidence on male-specific anatomical considerations.
  3. Organizational Flow Improvements:
    • Improve transitions between sections to ensure smoother flow and logical progression of ideas.
    • Use consistent formatting and subheadings throughout the manuscript to enhance readability.
  4. Incorporate recent literature:
    • Include references to enrich the discussion of pain management and complications. Suggested references include:
      • PMID: 24641263
      •  

 Description of Methods

  1. Sufficiency of Method Explanations:
    • Lines 407-437: Detail the measurement techniques for ultrasound imaging and the specific parameters used during the block.
    • Provide an overview of how anatomical variations were accounted for in the study.

Tables and Figures

  1. Recommendations for Improvements:
    • Figure 1: Include a more detailed flowchart of the steps involved in the lateral mid-shaft approach.
    • Table 1: Provide specific data on patient demographics (e.g., age, BMI range, and injury type).
    • Figure 2: Enhance readability with clearer labels and color coding.
  2. Titles, Descriptions, or Labels:
    • Ensure all figures and tables have descriptive captions that provide sufficient context for the data presented.
    • Table 2: Include detailed examples of the clinical outcomes achieved with this approach.

 Discussion

  1. Expand on the limitations of this technique, particularly in relation to:
    • Patient-specific anatomical challenges.
    • Situations where other approaches may still be preferable.
  2. Provide suggestions for future research, such as:
    • Investigating long-term outcomes of using this approach.
    • Comparing the effectiveness of this method with traditional approaches in a randomized clinical trial.

 Conclusion

  • Reiterate the practical benefits of the lateral mid-shaft approach while acknowledging the need for further validation through multicenter studies.

Reviewer 2 Report

Comments and Suggestions for Authors

I would like to congratulate the authors on developing a novel regional anesthesia technique for sciatic nerve block.

The purpose of this brief report was to describe in detail a novel lateral approach technique for sciatic nerve block to simplify the procedure for specific categories of patients and for less experienced practitioners.

This report brings several benefits to the existing techniques:

Strengths

Novelty: This report describes a novel ultrasound-guided lateral mid-shaft approach for sciatic nerve blocks, designed to overcome some existing difficulties in some patient populations, such as individuals with obesity

Significance: this technique improves patient comfort and simplifies execution, making it more accessible for less experienced practitioners;

Better visualization: The report includes detailed figures and diagrams that illustrate the anatomical structures and procedural steps improving comprehension.

Limitations

Lack of a comparative evidence: the report lacks a direct comparison with existing established methods including effectiveness, patient-related outcomes, and complication rates.

Methodology: the report is based on retrospective observations and lacks robust prospective comparison;

Data: Despite mentioning over 5,000 procedures, the reports does not provide any detailed data on patient demographics, BMI variations, or specific clinical situations;

Complication and success rates: risks such as intraneural or intravascular injections are discussed but not analyzed quantitatively and/or compared with other techniques.

Applicability: The performance of the reported technique across various clinical settings and with operators of differing expertize levels is not well addressed.

Recommendations for Improvement

-Comparison of success rates and complication rates with other established approaches would be important;

-Patient demographics and procedural details’

-Some quantitative data/analysis regarding the success/failure rates

Round 2

Reviewer 2 Report

Comments and Suggestions for Authors

Dear Authors,

Many thanks for addressing the comments.